# A Review of the Nutritional Approach and the Role of Dietary Components in Children with Autism Spectrum Disorders in Light of the Latest Scientific Research

**DOI:** 10.3390/nu15234852

**Published:** 2023-11-21

**Authors:** Seda Önal, Monika Sachadyn-Król, Małgorzata Kostecka

**Affiliations:** 1Department of Nutrition and Dietetics, Health Sciences Institute, Ankara University, 06110 Ankara, Turkey; dyt.sedaonal@gmail.com; 2Department of Nutrition and Dietetics, Faculty of Health Sciences, Fırat University, 23200 Elazığ, Turkey; 3Faculty of Food Science and Biotechnology, University of Life Sciences in Lublin, 20-950 Lublin, Poland; monika.sachadyn-krol@up.lublin.pl

**Keywords:** autism spectrum disorders, children, elimination diet, dietary component, eating disorders

## Abstract

Autism spectrum disorder (ASD) is a neurodevelopmental disorder that affects several areas of mental development. The onset of ASD occurs in the first few years of life, usually before the age of 3 years. Proper nutrition is important to ensure that an individual’s nutrient and energy requirements are met, and it can also have a moderating effect on the progression of the disorder. A systematic database search was conducted as a narrative review to determine whether nutrition and specific diets can potentially alter gastrointestinal symptoms and neurobehavioral disorders. Databases such as Science Direct, PubMed, Scopus, Web of Science (WoS), and Google Scholar were searched to find studies published between 2000 and September 2023 on the relationship between ASD, dietary approaches, and the role of dietary components. The review may indicate that despite extensive research into dietary interventions, there is a general lack of conclusive scientific data about the effect of therapeutic diets on ASD; therefore, no definitive recommendation can be made for any specific nutritional therapy as a standard treatment for ASD. An individualized dietary approach and the dietician’s role in the therapeutic team are very important elements of every therapy. Parents and caregivers should work with nutrition specialists, such as registered dietitians or healthcare providers, to design meal plans for autistic individuals, especially those who would like to implement an elimination diet.

## 1. Introduction

Autism spectrum disorder (ASD) is a multifaceted neurodevelopmental condition that is defined by the presence of core symptoms such as restricted/repetitive behaviors, language delay, and social interaction/communication impairment [1]. In addition to these definitions, an ASD diagnosis often co-occurs with other conditions, including motor abnormalities, gastrointestinal (GI) problems, epilepsy, intellectual disability, and sleep disorders [2]. ASD typically manifests before the age of 3 years and may persist throughout an individual’s lifetime. According to the Centers for Disease Control and Prevention (CDC) [3] report of March 2023, 1 in 36 children aged 8 years is diagnosed with ASD, marking an increase from 1 in 44 children in the previous reporting period. ASD is nearly four times more common among boys than girls [4].

The etiology of ASD is still poorly understood, and ASD is considered to be a multifactorial disorder caused by genetic, epigenetic, and environmental factors [1,5]. It is emphasized that genetic factors are responsible for only 10–20% of ASD cases, and according to the literature [6], nearly half of the symptoms associated with ASD may be due to environmental factors [7]. In the group of environmental factors, nutrition plays a critical role in the development of ASD [6].

The levels of glutathione (GSH), a primary intracellular antioxidant and detoxifying agent which acts as a methyl donor for many metabolic reactions, and S-adenosylmethionine (SAM) are significantly lower in ASD due to oxidative stress. Specific nutritional factors, including sulfur amino acids such as cysteine (CYS) and methionine (MET), as well as folate, vitamins B12, and B6, play a critical role in supporting the metabolic pathways responsible for GSH and SAM. These factors are of paramount significance in the dietary requirements of individuals with ASD. Mitochondria play a key role in oxidative stress and in maintaining the cellular redox state. Dietary supplements containing nearly all nutritional factors that support the metabolic pathways generating GSH and SAM have been shown to improve hepatic mitochondrial function in mice with diet-induced obesity [6,7,8].

Children with ASD experience more general GI symptoms, including abdominal pain, diarrhea, and constipation [9]. It has been reported that 83–91% of ASD individuals have co-occurrence of GI symptoms [10]. Children with ASD who are unable to express their emotions are known to manifest GI symptoms in extreme behaviors such as aggression, self-injury, and excessive self-repetitive behavior [11]. Given the prevalence of GI symptoms in ASD and other key behavioral areas, new research has been undertaken to explore the changes in the gut microbiota and the metabolites that may be associated with these symptoms [12]. In individuals with ASD, altered gut microbiota can influence the immune system and lead to the release of metabolites, which points to a connection between dysbiotic gut microbiota and ASD [13,14,15].

Due to the ambiguity of the causal relationship, the relative inefficacy of ASD treatment strategies, and insufficient knowledge about the role of dietary factors in the development of ASD, patients are currently being exposed to numerous “complementary and alternative interventions” by researchers and caregivers. Some of these interventions specifically target the diet and gut health [16,17,18]. Figure 1 summarizes the problems that may be associated with nutrition and nutritional therapies in ASD.

Do children with ASD eat differently? Do they have eating disorders that require dietary modification and separate dietary patterns? Can the described dietary modifications or elimination diets be applied to children with ASD and is such dietary management safe for them? To answer these questions, the main objective of this review was to describe the existing clinical and experimental knowledge and to determine whether nutrition and specific diets can potentially alter gastrointestinal symptoms and ASD.

## 2. Methods

The review was performed by a team of Polish and Turkish researchers as part of a broader study on the role of diet in alleviating ASD in Polish and Turkish children. The study involved a narrative literature review as a comprehensive, critical, and objective analysis of the current knowledge on the topic of ASD and dietary components. Electronic databases such as Science Direct, PubMed, Scopus, Web of Science, and Google Scholar were searched to find related studies published between 2000 and September 2023 (all available articles were analyzed in the case of the Feingold diet). A comprehensive search was performed using the following keywords: autism, autistic disorder, ASD, autism spectrum conditions, cross-sectional study, nutrition, elimination diet, restriction diet, and nutritional approach. All related studies were identified and transferred into the EndNote software (version 20.6) to select and manage the references. The reference lists of the related studies were also examined manually to find other potentially eligible studies. A total of 155 articles were ultimately selected for review.

## 3. Selective Eating as a Nutritional Problem

Selective eating is generally defined as restrictive eating behavior with a limited number of preferred foods. Selective eating is often associated with eating disorders, in particular when it has negative health or psychosocial consequences [19]. There is no consistent operational definition of selective eating, which is why the prevalence of selective eating is difficult to estimate in children both with and without ASD. According to current estimates, selective eating is more prevalent in children with ASD than in typically developing children, where it has been estimated at 15–20% [20]. A literature review conducted by Ledford and Gast (2006) demonstrated that behavioral feeding disorders affect 46–89% of children with ASD [21]. In the reviewed studies, selective eating was generally associated with frequent food refusals, a limited range of foods, high intake of a few acceptable foods, and selective preferences for some groups of food products. Children with ASD have not developed healthy eating skills, which is why they face challenges during mealtimes. In individuals with ASD, food preferences, aversions, and food refusal could be influenced by sensory sensitivity to textures, tastes, and smells [22]. Individuals with ASD often experience GI problems such as abdominal pain, constipation, and diarrhea, which could be related to selective eating [9]. Oral motor impairments and fine motor impairments in persons with ASD can influence swallowing, chewing, and utensil use, which further contributes to feeding difficulties [23]. Behavioral inflexibility and a need for sameness are some of the core challenges in ASD, and they prevent diverse eating experiences which are characterized by frequent changes in menu, utensils, dishes, and environments [24]. Mealtime is also a social activity that involves interaction and conversation. In ASD, the high prevalence of co-occurring anxiety increases the risk of food neophobia, namely the fear of trying new foods [25]. Children have to be repeatedly exposed to novel and potentially anxiety-inducing foods to develop healthy food preferences and eat a varied diet. Individuals with ASD often lack the cognitive skills for managing novelty, forming prototypes, and generalization, and they could find it difficult to process the similarities and differences in food groups, including differences in the color, flavor, and consistency of cheese. In the work of Yamane et al. (2020), children with food selectivity were classified based on the type of encountered difficulties (Table 1) [26].

In a recent study by Amin et al. (2022), individuals with ASD consumed 44 out of the 75 food items listed in the Food Preference Questionnaire (FPQ), and they were more selective than typically developing (TD) participants who ate 51 food items on average [27]. Similar observations were made in previous studies. Children with ASD consumed a smaller variety of foods than children without (22.8% versus 3.5%, *p* = 0.002). In the studied population, 35% of children with ASD, but only 3.5% of the control subjects, exhibited food selectivity toward starchy foods. Children with ASD most often refused to eat meat, eggs, rice, vegetables, and fruits [28]. In the group of 279 patients who were assessed over a period of 24 months, 70 children with ASD and severe food selectivity were eligible for inclusion in the study. According to the surveyed caregivers, vegetables were excluded by 67% and fruits were excluded by 27% of the studied population. In 78% of the children, the consumed diets increased the risk of five or more nutritional deficiencies [29]. Malhi et al. (2017) also found that children with ASD ate fewer foods, in particular fruits, vegetables, and proteins, than typically developing children [30]. Children with ASD refused more food items and were six times more likely to be picky eaters than the control subjects. The foods offered at home were also analyzed to rule out the influence of the family’s dietary choices on the food intake of children with ASD. Interestingly, no significant differences were reported in the number of consumed foods from each food group, or the total number of foods consumed by families with and without autistic children. In addition, the diets consumed by the parents of children with ASD were significantly more varied than the children’s diets, which clearly indicates that food selectivity in children with ASD did not result from food restrictions in the family. Postorino et al. (2015) evaluated the clinical and behavioral characteristics of autistic children aged 3–11 years and compared children with and without eating problems, including GI symptoms, food refusal, and food selectivity [31]. The severity of ASD symptoms was assessed by both professionals (ADOS-G) and parents (Autism Diagnostic Interview-Revised—ADI-R, Social Responsiveness Scale—SRS, Social Communication Questionnaire—SCQ). Interestingly, the prevalence of ASD symptoms in children with food selectivity was significantly higher in parental assessments than in professional assessments. Similar findings were reported by Allen et al. [32], Zachor and Ben-Itzchak [33], and Prosperi [34]. According to Saban-Bezalel et al. (2021), this discrepancy could be attributed to the continuous impact of atypical eating habits throughout childhood, which can influence parental perceptions of symptom severity. In turn, professionals are generally not familiar with the long-term eating habits of the assessed children [35]. 

Restrictive food intake in a selective diet can lead to nutritional deficiencies and pose a potential health risk associated with feeding problems. However, research investigating nutrient intake in autistic children produced contradictory findings. According to a meta-analysis of 56 studies conducted by Bourne et al. (2022), low body weight or substantial weight loss are the most noticeable outcomes of a severely limited diet, especially in children where a restrictive diet may undermine the achievement of the expected growth and developmental milestones [36]. Despite the above, Avoidant/Restrictive Food Intake Disorder (ARFID) is not always correlated with low weight. In some cases, autistic children and adolescents were overweight due to excessive consumption of a narrow range of energy-dense foods high in fat, sugar, or salt. Low dietary variety can have significant consequences, and it can deprive individuals of essential nutrients such as iron and vitamins [37,38]. The results of recent studies investigating the nutrient intake of children with ASD are presented in Table 2. A meta-analysis of 17 prospective controlled trials revealed that protein and calcium intake was considerably lower in children with ASD than in typically developing children [29]. Several researchers also reported a higher prevalence of iron deficiency and anemia among children with ASD [39]. In other studies, healthy children were characterized by significantly higher average levels of hemoglobin, ferritin, magnesium, potassium, calcium, phosphorus, glucose, and hematocrit than autistic children [40]. 

## 4. Nutritional Approaches 

### 4.1. Diets 

#### 4.1.1. Gluten-Free and Casein-Free Diets

High levels of urinary peptides have been identified in children with ASD [45,46,47]. Many of these peptides can be classified as exorphins (exogenous opioids), including casomorphins and gluteomorphin derived from dietary sources containing gluten and casein. This observation provides further evidence for incomplete protein digestion and increased intestinal permeability in individuals with ASD [48,49]. Circulating peptides can cross the blood–brain barrier, directly influence the central nervous system, and exert adverse effects on attention, brain development, social communication, and learning. [50,51].

It has been hypothesized that a diet low in these proteins may normalize urinary peptide levels and improve the behavioral symptoms of affected children. Several recent studies analyzing the effects of dietary treatments on ASD [52,53,54,55,56,57] are presented in Appendix A. In most studies examining the impact of a GFCF diet on autistic children, the studied population is usually small, and the diet is applied for a short period of time. Prolonged adherence to a GFCF diet could lead to nutrient deficiencies, social isolation of children with ASD (similarly to other restrictive diets), and considerable economic burden for families. Therefore, a GFCF diet is not fully beneficial for children with ASD.

#### 4.1.2. Ketogenic Diet 

The ketogenic diet (KD) is high in fat, extremely low in carbohydrates, and low in proteins. Individuals who adhere to the KD obtain 90% of dietary energy from fat, 7% from protein, and 3% from carbohydrates [58]. In individuals with ASD, KD may improve social behavior by normalizing GABA, enhancing mitochondrial function, reducing inflammation and oxidative stress in the brain, inhibiting the mTOR signaling pathway, and modulating the gut microbiota. Researchers analyzing the effects of the KD did not observe significant behavioral improvements in animal models of ASD, and preliminary human studies demonstrated that KD was effective in improving the core symptoms of ASD [59,60] (Table 3). The efficiency of the KD must be monitored with the use of urinary ketones and serum beta-hydroxybutyrate (BHB).

It should be noted that despite encouraging results, human studies examining the applicability of the KD in subjects with ASD have several important limitations. These limitations include smaller sample sizes, difficulty in adhering to the KD, discrepancies between the duration and composition of the KD, high dropout rates resulting from the unpalatability of the KD, and nutritional deficits. In addition, the KD is associated with a higher risk of inflammation and mitochondrial dysfunction, as well as adverse effects such as constipation, reflux, and other comorbidities [7]. The side-effects of the KD described in the literature are presented in Table 4. The KD seems effective in ASD patients, but the reviewed clinical studies had small sample sizes, probably because randomized trials are difficult to set up for children with ASD. Additional studies are warranted to understand the effects of the KD in individuals with ASD [67]. 

#### 4.1.3. Feingold Diet

In addition to the most common elimination diets, various dietary modifications have been sought to treat children with ASD. One such modification was polarized in the 1970s by the controversial Feingold diet. This dietary regime involves the elimination of artificial food dyes, flavors, sweeteners, preservatives, as well as foods containing butylated hydroxyanisole (BHA), butylated hydroxytoluene (BHT), tert-butylhydroquinone (TBHQ), and salicylates [76]. Despite considerable success, the Feingold diet has been widely criticized by the medical community for a lack of scientific evidence, strict rules, and potential health risks [77]. In later years, several controlled studies were conducted to compare the behavior of hyperactive children on the Feingold diet and a placebo diet (diet crossover designs), or to examine specific behavioral responses to the presence of food dyes in the diet [78]. In one study, autistic children aged 3–6 years old were placed on the Feingold diet coupled with a language training program. Non-significant differences in language education performance and the development of receptive language skills were observed before and after the intervention [79]. Several reviews [80,81,82] and one meta-analysis [83] have concluded that the Feingold diet is not an effective treatment for hyperactivity. Schab and Trinh (2004) conducted a meta-analysis focusing on the effect of artificial food dyes on hyperactivity rather than the Feingold diet as a whole [84]. Their results corroborate the hypothesis that artificial food dyes can increase hyperactivity in some children. On the other hand, reduced phenol sulfotransferase (PST) activity in ASD individuals compared to controls [85] points to the feasibility of the Feingold diet.

#### 4.1.4. Candida Diet

The influence of *Candida* spp. on the immune system, brain, and behavior of children with ASD has been examined by many researchers. Numerous studies demonstrated that *Candida* spp. were isolated more frequently from the stool samples of autistic patients than healthy subjects. Doreswamy et al. (2020) found much higher yeast counts in the autistic group than the control group [86]. Iovene et al. (2017) also identified aggressive forms (pseudohyphae) of *Candida* spp. in stool samples from 57% of children with ASD [87] In the work of Emam et al. [88], the *Candida* spp. counts were also much higher in individuals with ASD (81.9%) than in healthy controls (28%), but the observed differences were not statistically significant [89,90]. In some studies, yeasts were often identified under a microscope, but they were rarely observed in samples collected from autistic or control groups, which points to the high ambiguity of the reported results [91].

The following *Candida* species were most frequently isolated from the stool samples of children with diagnosed or suspected ASD: *C. albicans* (57.4%), fluconazole-resistant *C. krusei* (19.8%), and *C. glabrata* (14.8%) [92,93]. None of these pathogens were identified in healthy children [94]. Most studies demonstrated that increased counts of *Candida* spp. did not affect the severity of symptoms in autistic children and were not correlated with GI symptoms [87,92]. Research conducted in the 1980s suggested that yeasts play a significant role in behavioral and learning problems in children with ASD [95]. It was noted that excessive and long-term exposure to antibiotics (for example, in the treatment of middle ear infection) led to gut dysbiosis and caused the overgrowth of *C. albicans* yeast in the intestinal tract. Srikantha and Mohajeri (2019) reported that *Candida* spp. may increase serotonin production in the intestine (5-hydroxytryptamine, 5-HT) and reduce serotonin synthesis in the brain (due to the consumption tryptophan, a precursor of serotonin), which can lead to hyperserotonemia and behavioral outcomes in ASD children [96].

Autistic children are picky eaters, and their diet usually consists of a very narrow range of foods, depending on type, texture, or appearance [97]. These children have a preference for starchy and fatty foods, simple carbohydrates, snacks, and processed foods [98]. Diets abundant in carbohydrates, such as glucose and mannose [99], correlated positively with the abundance of gut *Candida* [100], whereas negative correlations were observed between high-protein diets and the abundance of fungi in healthy volunteers [101]. Furthermore, mannitol, sorbitol, xylose, adonitol, and xylitol were also found to significantly promote the growth of *C. albicans* (100–200%), whereas other metabolites, such as raffinose, arabinose, trehalose, lactose, galactinol, galactitol, and arabitol, had a low or marginal impact on the proliferation of *C. albicans* [100]. Added sugars (including honey, jam, and candy), highly refined carbohydrates (in particular, flour-based products), red and cured meats, and dairy products should be avoided to prevent the growth of *Candida* spp. [99]. Anti-Candida diets have been recommended for autistic children due to their health benefits [102]; however, there is no conclusive empirical evidence to indicate that these diets inhibit *Candida* growth or promote gut health. The information about the efficacy of these diets is insufficient to make recommendations for their use. 

Probiotics appear to be a promising treatment for reducing GI disturbances and the overgrowth of *Candida* in children with ASD [103,104]; however, the efficacy of probiotics remains controversial and requires further research. Probiotics such as *Sacharomyces boulardii* [102] and *Lactobacillus* spp. [104] may induce changes in the intestinal microflora and inhibit the growth of pathogens by stimulating the production of β-defensin and IgA. The proliferation of *Candida* can be suppressed by IL-17 and IL-22, which are modulated by selected *Lactobacillus* species via tryptophan-derived aryl hydrocarbon receptor ligands [105]. Probiotics may strengthen the intestinal barrier by maintaining tight junctions and triggering mucin production. 

The composition of the gut microbiota could be modulated to minimize *Candida* overgrowth, GI problems, and ASD symptoms; however, human studies have provided limited or inconclusive evidence that a sugar-free diet or probiotic supplementation deliver beneficial effects in ASD. The current findings do not promote the use of these modalities in the treatment of ASD or the introduction of such changes into the patients’ diets.

#### 4.1.5. Specific Carbohydrate Diet

This diet (SCD) was developed by Dr. Sidney Haas for the treatment of celiac disease. This diet was developed on the assumption that carbohydrates (sugars) feed bacteria and yeasts in the intestines, thus leading to an over-abundance of bacteria and yeast [106]. As a result, carbohydrates remain undigested in the intestines, which provides even more food for bacterial and yeast growth.

Many autistic children have severe GI symptoms, including diarrhea, constipation, bloating, and pain [107]. The associated functional GI abnormalities include low levels of disaccharidase enzymes [108,109] and defective sulfation of ingested phenolic amines, such as acetaminophen [110]. Some ASD specialists believe these symptoms could be caused by bacterial or fungal overgrowth in the intestines [91,111]. The SCD eliminates complex starches that feed bacteria and yeasts in the intestines, which improves ASD symptoms and mitigates functional GI abnormalities. 

In the SCD, foods are described as “legal” or “illegal” based on their carbohydrate content. “Illegal” carbohydrates include all cereals, (maize, wheat, wheat germ, barley, oats, and rice), sugars, beans, potatoes, and all processed foods (including canned and preserved vegetables). Milk and milk products with a high lactose content, ice cream, sweets, chocolate, and products containing fructooligosaccharides (FOS) are also placed on the list of prohibited items. Seaweed and related products should also be eliminated from the diet. “Legal” carbohydrates include unprocessed meats, vegetables, fruits, and some dairy products (a casein-free version of this diet has also been proposed). Certain legumes, including dried navy beans, lentils, peas, split peas, unroasted cashews, shelled peanuts, all-natural peanut butter, and lima beans, are also allowed. Most oils, teas, coffee, mustard, apple or white vinegar, and juices without additives or sugars can be a component of a varied diet, and honey can be used as a sweetener in special cases [112,113].

The SCD is already naturally gluten-free. In a case study conducted by Barnhill et al. (2020), the SCD led to a considerable increase in protein intake which exceeded the current RDA levels [110]. However, high-protein diets have been found to increase the intestinal absorption of calcium, increase the levels of circulating insulin-like growth factor-1, and decrease the serum parathyroid hormone level [114]. In a recent study, the intake of higher animal protein, calcium, and phosphorus was positively associated with bone density measures in autistic children, and the authors concluded that children with ASD should focus on higher protein intakes than the RDA [115].

Despite the fact that the effectiveness and safety of the SCD protocol for autistic children with GI problems has not been evaluated to date, this diet is widely used by many families, with or without clinical guidance [110]. Restrictive dietary interventions may lead to nutritional deficiencies, especially when not clinically supervised or when the patients exhibit selective eating patterns and restricted dietary diversity. These observations indicate that the use of the SCD protocol in patients with ASD warrants additional investigation.

### 4.2. Supplements

In the absence of curative treatments for ASD, and due to changes in the patients’ eating behavior, nutritional supplements and alternative medicine solutions have been extensively promoted among the families of individuals with ASD. Despite a relatively large number of studies on this subject, many myths and uncertainties still persist. The use of nutritional supplements, including omega-3 fatty acids and multivitamins, lacks support from current scientific evidence and should not be recommended as the official guidelines [116]. 

#### 4.2.1. Vitamins and Minerals

Multivitamin and mineral supplements: Some findings suggest that multivitamin and mineral supplements may improve sleep and resolve digestive issues in autistic children. Additionally, such supplements can provide essential nutrients that may be lacking or insufficient in their diets (see Table 2). However, before starting any supplementation, the patients should consult a doctor or a dietitian to ensure that the supplements are safe and appropriate.

The evidence available for most vitamin and mineral supplements is insufficient to endorse a therapeutic supplementation approach to ASD. The latest evidence on the impact of supplements on ASD symptoms and health parameters is presented in Table 5. The scientific evidence for certain dietary components is outdated and often controversial. For instance, there is a lack of recent experiments concerning vitamin C. The application of vitamin C in addressing ASD has not gained widespread traction as a therapeutic approach. An initial study assessing the impact of a moderate dose of multivitamin and mineral supplements on children with ASD concluded that autistic children are deficient in vitamin C and that supplementation leads to clinical improvement [110]. However, the existing research has numerous limitations. A considerable portion of supplementation research is currently in the animal testing phase. For example, some studies revealed that selenium supplementation increased selenium levels, led to substantial improvement in social interactions and cognitive function, decreased repetitive stereotypical behaviors, and altered neurotransmitter levels in BTBR mice, which are the most widely used animal model for ASD research. The authors concluded that selenium exerts a potentially protective effect on the hippocampus of BTBR mice by regulating the neurotransmitter levels, reducing oxidative stress, and mitigating neuroinflammatory responses and neural cell injury [117]. The impact of supplementation on pregnant women and the risk of ASD has been studied in recent years. Insufficient nutrient intake during pregnancy is linked with various adverse outcomes, including elevated risk of atypical behavior and neuropsychiatric conditions such as depression, anxiety, schizophrenia, ASD, and ADHD, as well as impaired cognition, visual problems, and motor deficits [118]. The use of iodine supplementation during pregnancy, particularly when initiated prior to conception or during the first trimester, was shown to be more effective in preventing neurological damage.

#### 4.2.2. Omega-3 Fatty Acids

Omega-3 long-chain polyunsaturated fatty acid (*n*-3 LCPUFAs) supplements are highly popular in the treatment of ASD (Table 5). These fatty acids and their metabolites have been linked with ASD because they significantly contribute to brain structure and function, neurotransmission, cell membrane composition, and organization within microdomains, and play important roles in inflammation, immunity, and oxidative stress [133,139]. Recent research has shown that omega-3 fatty acids have the potential to improve social communication with a small effect size, but these conclusions were based on very low-quality evidence from studies involving children and adolescents [140]. According to Horvath et al. (2017), omega-3 fatty acid supplementation does not improve the outcomes for children with ASD [141].

#### 4.2.3. Probiotics and Prebiotics

Gastrointestinal symptoms such as diarrhea, constipation, and alternating diarrhea/constipation are very common in ASD and are associated with the severity of the neurobehavioral disorder. There is increasing evidence to suggest that disturbances in the pathway underlying the microbiota–gut–brain axis, in particular dysregulated gut microbiota, can cause neurobehavioral and gut dysfunction in patients with ASD [142]. Numerous researchers have analyzed animal models of ASD, in particular BTBR mice that exhibit symptoms of ASD. Mouse models may be helpful in assessing the impact of gut dysbiosis by presenting the correlations between ASD traits and their severity vs. gut microbiota. Correti et al. [143] analyzed an animal model to determine the impact of different microbiota profiles on gut status and permeability, and ASD-specific behaviors. The study revealed definite gender differences in the gut microbiome, which also influenced behavioral variation. These results are promising, and they indicate that animal models of ASD may contribute to a better understanding of the mechanisms underlying the gut–microbiota–brain axis in humans.

The human GI tract hosts around 100 trillion bacteria that roughly represent 1000 different species. In a healthy adult GI tract, Bacteroidetes and Firmicutes are the predominant bacterial phyla that collectively represent 70–90% of total gut bacteria [144]. Previous studies have shown that ASD patients are characterized by a higher Firmicutes:Bacteroidetes ratio [45]. Microorganisms such as bacteria, viruses, protozoa, fungi, and archaea that colonize the human gut are collectively referred to as microbiota [10]. In turn, the microbiome is defined as the collective genomes of microorganisms in a particular environment [145]. The gut microbiome enters into symbiotic interactions with various bodily systems and organs [146], and the gut microbiota play a very important role by extracting energy and nutrients from food. As regards the immune system, gut microbiota protect the host against external pathogens by secreting antimicrobial substances and promoting the development of intestinal mucosa and the immune system [147]. Moreover, gut microbiota may regulate the central nervous system (CNS) via endocrine, neural, and immune pathways [142].

The potential of probiotic and prebiotic supplements for improving gut health has been researched extensively in recent years. Probiotics are live microorganisms which provide health benefits to the host when taken in adequate amounts [145]. Prebiotics also play an important role in intestinal health [148]. Prebiotics are food ingredients that are metabolized by beneficial indigenous bacteria and positively modulate the gut microbiota [149]. Probiotics stabilize the mucosal barrier [150], which has led some scientists to conclude that probiotics can improve GI health and balance behavioral abnormalities in individuals with ASD [144]. Furthermore, the modification of the gut microbiota via the use of probiotics, prebiotics, synbiotics, or fecal microbiota transplantation can influence the symptoms or progression of ASD via microbiota-mediated immune responses, endocrine pathways, and direct neural mechanisms [151]. Selected studies [49,123,149,150,152] examining the role of prebiotics and probiotics in ASD treatment are summarized in Appendix A. 

Despite evidence which suggests that prebiotics and probiotics can potentially reduce the symptoms and severity of ASD, more studies are needed to confirm their efficacy in treatment. 

### 4.3. Strengths and Limitations

The main strength of the presented review is that it is based on the results of the latest clinical and experimental studies and summarizes the current knowledge on the impact of different dietary models on the behavior and health of individuals with ASD. The main weaknesses of the reviewed studies include the lack of control groups, the parents’ and physicians’ high expectations regarding the benefits of the interventions, which can affect the results of nutrition studies, as well as the general scarcity of blinded studies to reduce bias. Additionally, the reviewed studies differed considerably in design, participants, duration, and measurement techniques, which may ultimately affect the results and the formulated conclusions. Researchers often study the effects of dietary therapies on different aspects of ASD and rarely compare their effectiveness with other interventions, which undermines the reliability of generalizations and comparisons between therapies. In addition, factors such as adherence to a dietary intervention or the use of other treatments than a dietary intervention (drug treatment, education, etc.) may produce variable effects in different groups of patients with ASD. Another weakness of the study presented is that a systematic search process has not been followed and therefore the review is not replicable. The quality of the studies used for the synthesis of results has not been assessed. Neither publication bias nor selection bias can be ruled out.

## 5. Conclusions

Autism spectrum disorder (ASD) is a heterogeneous condition, which implies that individuals with ASD are characterized by a wide range of behavioral symptoms, as well as co-occurring medical and mental health issues. As a result, no diet or supplementation regime can be tailored to the specific combination of ASD symptoms in each individual. In turn, a one-size-fits-all dietary approach does not take into account all individual ailments. In 2006, the UC Davis MIND Institute launched the Autism Phenome Project (APP) to identify clinically relevant, homogeneous subtypes of ASD to improve our understanding of its etiology and, above all, to promote the search for targeted treatments and dietary interventions [153]. Sensory behaviors, behavioral disorders, level of cognitive development, or anxiety disorders may be the factors by which to group autistic patients in relation to dietary recommendations, and they facilitate nutritional interventions combined with supplementation. Research has shown that children with ASD eat differently regardless of the progression of the disease and the classification of symptoms. Many of them have eating disorders that require dietary modifications and different eating patterns. It remains unclear whether nutrition and specific diets can potentially alter gastrointestinal symptoms and neurobehavioral disorders in children with ASD. The presented results pave the way to further research and give hope that a better understanding of ASD will enable the development of effective nutritional strategies. At present, there is no cure for ASD, and treatment focuses on nutritional and behavioral therapies to improve the patients’ social functioning. Gastrointestinal disorders frequently accompany ASD and are often used to classify patients with ASD, which is why nutritional interventions may play a significant role in alleviating GI symptoms. Despite the fact that various dietary interventions have been studied, there is a general lack of conclusive evidence that dietary therapies are safe and effective in the treatment of ASD, especially in individuals who adhere to significant dietary restrictions and modify/restrict their intake of major dietary components. Based on the current knowledge, it appears that no definitive recommendations for a specific nutritional therapy can be made as a standard treatment for ASD. However, many parents and healthcare providers opt for dietary interventions and report promising results in terms of reduced behavioral disorders and GI symptoms.

An individualized dietary approach and the dietitian’s role in the therapeutic team are very important elements of every therapy. These issues will be addressed in our upcoming study on nutritional support based on recommendations and guidelines.

## Figures and Tables

**Figure 1 nutrients-15-04852-f001:**
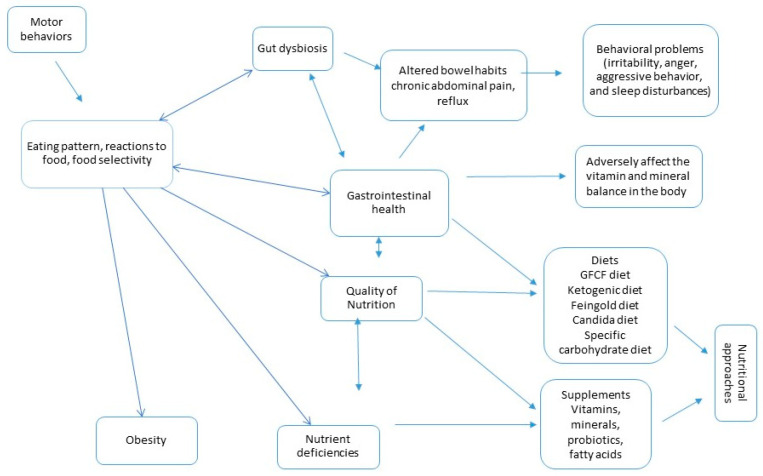
Problems that may be related to nutrition in ASD and current nutritional therapy approaches.

**Table 1 nutrients-15-04852-t001:** The type and the developmental characteristics of autistic children with selective eating habits according to the classification proposed by Yamane [26].

	Tendency	Brief Characteristics
Group 1 (sensory)	Foods are selected based on sensory factors, such as texture, smell, taste, temperature, and color	Severe intellectual disability (the sensory motor stage in the theory of mental development) • Oral hypersensitivity
Group 2 (visual)	Foods are judged at a glance based on color, shape, or cooking method	Moderate intellectual disability • Severely restricted imagination
Group 3 (familiarity)	Selection of familiar foods	Visual performance is dominant • Developmental age for cognition >2 years • Difficulty in social imagination (prediction)
Group 4 (environmental stimulation)	Food choices are affected by environmental factors: place, plate, cup, people, and room temperature	Attention deficit disorder • Each developmental age is variable

**Table 2 nutrients-15-04852-t002:** Nutrient intake in children with ASD.

Nutrient	Intake	Subjects	Source
Protein Other Macronutrients, Vitamin A Vitamin B1, B2, B3, B6 Folic Acid Vitamin B12 Calcium Iron Zinc Selenium	Lower in ASD No difference Lower in ASD Lower in ASD Lower in ASD Lower in ASD Lower in ASD Lower in ASD Lower in ASD Higher in ASD	86 children with ASD aged 2–8 years and 57 age-matched peers without ASD	Barnhill et al. [41]
Energy Fats Potassium Copper Folate Iron Vitamin C	No difference No difference Lower in ASD Lower in ASD Lower in ASD Lower in ASD Lower in ASD	63 ASD children in the age range of 4 to 10 years and 50 typically developing children matched on age and socio-economic status to the ASD children	Malhi et al. [30]
Protein Calcium Iron Zinc Vitamin B2	Adequate (RDA) Lower than RDA Lower than RDA Lower than RDA Lower than RDA	53 children with ASD (45 boys and 8 girls) in the age group of 2–13 years	Siddiqi et al. [42]
Fiber Calcium Vitamin E Vitamin A Vitamin C Folic Acid Zinc	Lower than DRI Lower than DRI Lower than DRI Lower than DRI Lower than DRI Lower than DRI Lower than DRI	70 children with ASD and severe food selectivity	Sharp et al. [29]
Folate Zinc Vitamin B12 Vitamin D	Lower in ASD Lower in ASD Lower in ASD Lower in ASD	738 ASD children and 302 typically developing children (TD) age 2–6	Zhu et al. [43]
Vitamin E Vitamin K Vitamin B_2_ Vitamin B_6_ Vitamin A Vitamin D Vitamin B12 Folate Magnesium Phosphorus Zinc Selenium	Lower in ASD Lower in ASD Lower in ASD Lower in ASD Lower than DRI in ASD & TD Lower than DRI in ASD & TD Lower than DRI in ASD &TD Lower than DRI in ASD & TD Lower than DRI in ASD & TD Lower than DRI in ASD & TD Lower than DRI in ASD & TD Lower than DRI in ASD &TD	52 ASD cases (37 boys and 15 girls), 51 TD children (26 boys and 25 girls) aged 3–6	Alkhalidy et al. [44]

RDA—Recommended Dietary Allowance; DRI—Dietary Reference Intake.

**Table 3 nutrients-15-04852-t003:** The ketogenic diet as potential treatment for ASD.

Study Group	Scientific Evidence	References
	**Animal Studies**	
The study analyzed the protective effects of a KD on sociability, spatial learning, memory, and electroencephalogram seizures in glut3 heterozygous null (glut3+/−) mice exhibiting features relevant to ASD.	The study reported on a KD-related partial restoration of social features and alleviation of seizure events in male subjects. The KD also exerted neuroprotective effects in female subjects due to higher circulating and cerebrospinal fluid ketone concentrations and/or lower brain Glut3 concentrations.	Dai et al. [61]
The study examined mutant EL mice with comorbid epilepsy and ASD symptoms.	A clear sex-related difference was reported in response to the KD. The KD improved multiple measures of sociability and reduced repetitive behavior in female mice, but had limited effects in males.	Ruskin et al. [62]
The experiments involved En2 knockout mice exposed to a KD from postnatal day 21 to 60.	The early timing of a dietary intervention was recognized as an important factor in diet-dependent brain reorganization and maturation. Although monoamine levels in the forebrain regions were not affected in two null mice (En2(−/−)), increased social contact and reduced grooming behavior were evident in response to KD intervention.	Verpeut et al. [63]
The study analyzed sociability and social behavior in male and female MIA mice	Male MIA offspring were significantly asocial in the three-chamber sociability test, while female mice displayed normal and social behavior. After 3–4 weeks of KD treatment, the lack of sociability in male offspring was completely reversed and MIA-induced, self-directed, repetitive behavior was reduced.	Ruskin et al. [62]
	Human studies	
45 children aged 3–8 years	A KD reduced autistic manifestations in the Autism Treatment Evaluation Test (ATEC) and the Childhood Autism Rating Scale (CARS), in particular by improving sociability.	El-Rashidy et al. [64]
15 children aged 2–17 years	A modified ketogenic gluten-free diet supplemented using medium-chain triglycerides (MCTs) improved the social affect subdomain and scores in the total autism diagnostic observation schedule, 2nd edition (ADOS-2) scores, but it had no effect on restricted and repetitive behavior scores.	Lee et al. [54]
Six ASD (aged 4–14 years old) patients with a pathological increase in beta-hydroxybutyrate	A KD improved social communication in one of the six ASD patients and reduced the prevalence of comorbidities in patients, including attention deficit hyperactivity disorder (ADHD), compulsive behavior, preoccupation with parts of objects, and abnormal sleep.	Spilioti et al. [65]
6-year-old child	In a case study of a child with ASD, a KD improved behavior and intellect, and decreased the 18F-FDG uptake in the whole cortex.	Żarnowska et al. [66]

**Table 4 nutrients-15-04852-t004:** Why a ketogenic diet is not the best diet for ASD.

Most Common Side Effects	References
The KD can be difficult to maintain, especially in children with limited food preferences. It is important to have a plan in place to ensure that the child is able to stick to the diet. ASD patients also consume fewer foods and exhibit more feeding problems and diverse eating behaviors (selective intake, food refusal, food aversion, and atypical eating). Some foods are refused due to presentation or the need to use certain utensils. In a study by Albers et al. [68] 73% of the respondents rated adherence to the KD as more difficult, compared with age-matched controls, whereas only 26% of the subjects did not report such difficulties. These results confirm that the administration of a KD to ASD children is difficult.	Li et al. [69] Mayes & Zickgraf [70]
The sensory abnormalities commonly associated with ASD can influence the administration of and adherence to the KD. Parents reported that children with ASD were significantly more averse to food textures (*p* < 0.0001), in particular foods with a slimy and creamy texture. According to the authors, taste preferences and consistent food routines are important or very important determinants of the successful implementation of the KD.	Albers et al. [68] Balasco et al. [71] Cermak et al. [72]
Taste, smell, and texture hypersensitivities/aversions were regarded as the key difficulties in the implementation of a KD	Albers et al. [68]
Nutrient Deficiencies: The KD is very restrictive, and it may not provide growing children with the necessary nutrients. In children, the KD may suppress physical development and cause height deceleration. Parents should work with healthcare providers or dietitians to eliminate the risk of nutritional deficits.	Spulber et al. [73]
Possible Side-Effects: Some children may experience side effects from the KD, such as constipation, nausea, and vomiting. During the initial phase of the diet, common side effects also include hypoglycemia, metabolic acidosis, and refusal to eat.	Neal et al. [74] Newmaster et al. [75]

**Table 5 nutrients-15-04852-t005:** Recent research studies investigating the impact of dietary supplementation on ASD symptoms.

Supplement	Intervention	Subject	Main Results	References
Mg+B6	Mg: 50 mg for children aged 2–3 years, 100 mg for children aged 4–8 years, 200 mg for children aged 9–12 years. Vitamin B6: 25 mg for children aged 2–3 years, 50 mg for children aged 4–8 years and 100 mg for children aged 9–12 years. Duration of intervention: 3 months	70 children with ASD	The improvement in the overall score of the treated group was statistically significant relative to the placebo group and the intervention group. The improvement in the cognition and emotion score was statistically significant. The improvement in the social, communication and sensory deficiency score was not statistically significant.	Khan et al. [119]. A randomized, double-blind, placebo controlled study
Fe	3 mg/kg/day of liquid ferrous sulfate for 3 months	20 children with ASD	An improvement in iron levels and a reduction in the overall severity score on the Sleep Clinical Global Impression Scale were observed. Actigraphy measurements did not reveal significant improvements in the primary outcome measure, i.e., sleep onset latency and wake time after sleep onset.	Reynolds et al. [120]. A randomized placebo-controlled trial
Fe	Infusion of ferrous carboxymaltose (FCM) at 15 mg/kg up to a maximum dose of 750 mg	19 children with ASD (age: 4–11 years)	In most children (84.2%) exhibiting ASD, symptoms of restless legs, and serum ferritin levels below 30 μg/L experienced clinical amelioration and notably enhanced serum iron parameters following a sole intravenous ferric carboxymaltose (FCM) infusion.	DelRosso et al. [121]. Retrospective study
Se	Intervention group 1 received powdered selenium supplement at 1 × 20 g/day; intervention group 2 received a functional food product with a high content of selenium (bovine heart extract) at 50 g/day	65 children with ASD (age: 2–6 years)	Supplementation did not induce significant differences in total glutathione peroxidase levels.	Triana et al. [122] Randomized controlled trial
Zn	A dietary nutraceutical formula containing Zn was administered for 12 weeks. The daily Zn dose was adjusted to the participants’ body weight in kg plus 15–20 mg.	30 children with ASD (age: 3–8 years)	Zn supplementation markedly reduced CARS scores in children with ASD. Serum Zn and metallothionein levels increased significantly after Zn supplementation.	Meguid et al. [123]
Vitamin A	Participants with low plasma retinol levels (<1.05 μmol/L) received a single oral dose of vitamin A at 200,000 IU and completed a 6-month follow-up study	64 children (aged 1–8 years) with ASD	Vitamin A induced changes in the composition of gut microbiota and improved selected ASD-related biomarkers such as CD38 and RORA. In children with ASD, significant taxonomic associations with vitamin A were identified in the Bacteroidetes/Bacteroidales group.	Liu et al. [124] Single-blinded, non-randomized intervention pilot study
Vitamin A	200,000 IU for 6 months	33 ASD patients (mean age: 5.14 ± 1.33 years)	Vitamin A decreased serum levels of 5-hydroxytryptamine (5-HT). Differences in CARS scores were noted before and after the administration of vitamin A. Improvements were observed in social interactions, emotional responses, motor skills, adaptability, sensory sensitivity (including taste, smell, and touch), anxiety levels, and verbal and non-verbal communication. Moreover, the general impression and the overall score increased significantly after vitamin A supplementation. Furthermore, excluding restricted interests, all symptoms related to the neurodevelopmental deficits reported by parents improved substantially after vitamin A administration.	Guo et al. [125]
Vitamin B6	5 mg of vitamin B6/kg body weight per day for 2 weeks, followed by 10 mg of vitamin B6/kg body weight per day for 2 weeks. Total duration of treatment: 4 weeks	17 children with ASD (mean age: 8.8 years)	Five clusters were identified. Cluster 1 consisted of all persons who responded to vitamin B6. Persons with ASD may be highly heterogeneous.	Obara et al. [126] Single-arm intervention
Vitamin B6	No data	236 children with ASD (age: 3–16 years)	Supplementation with vitamin B and magnesium induced significant changes in tryptophan levels.	Kałużna-Czaplińska, et al. [127].
Vitamin B12	75 μg/kg every third day for 8 weeks	57 children with ASD (age: 3–7 years)	Vitamin B12 improved the CGI-I score, but not ABC or SRS. The supplement increased plasma methionine levels, decreased SAH levels, and improved the SAM-to-SAH ratio.	Hendren et al. [128]. Randomized controlled trial
Vitamin cB12 replaced with mB12; hB12		Four children with ASD	Anemia and metabolic acidosis showed improvement when cyanocobalamin (cB12) was replaced with methylcobalamin (mB12). Furthermore, homocysteine levels returned to normal after oral administration of 10 mg of hydroxocobalamin (hB12) in a single pediatric case.	Nashabat et al. [129].
Folic acid	400 μg, twice daily for 3 months	66 children with ASD (age: 4.5 ± 1.1 years)	Folic acid improved social engagement, cognitive language and preverbal abilities, receptive communication skills, emotional expression, and interactions and communication in children with ASD. In addition, positive changes in folic acid and homocysteine levels stabilized the glutathione-dependent redox balance.	Sun et al. [130]. Open-label trial
Folic acid	600 μg twice daily for 3 months Additional supplements (omega-3 and omega-6 fatty acids, carnitine) were administered for 12 months	67 children and adults with ASD (age: 3–58 years)	In a clinical assessment with blinding, folic acid improved non-verbal cognitive aptitude relative to the control (untreated) group. In a semi-blinded trial, the treated cohort demonstrated greater progress in ASD symptomatology and developmental progress than the control group. The concentrations of EPA, DHA, carnitine, vitamins A, B2, B5, B6, and B12, folic acid, and coenzyme Q10 increased markedly in the treated group and differed significantly from the control.	Adams et al. [52]. Single-blinded study
Vitamin D	300 IU/kg/day (max. 5000 IU/day) for 3 months	100 children with ASD (age: 6–9 years)	Vitamin D supplementation significantly alleviated the clinical symptoms of ASD measured on the CARS and ATEC scales. Supplementation did not induce significant changes in the serum levels of serotonin and IL-6 on the ABC-C scale.	Moradi et al. [131]. Randomized controlled trial
Vitamin D	300 IU/kg/day (max. 6000 IU/day) for 15 weeks	A total of 43 children, including 22 children with ASD (age: 3–13 years)	Vitamin D supplementation induced a significant decrease in irritability and hyperactivity on ABC-C subscales. Supplementation decreased lethargy/social withdrawal, inappropriate speech, and stereotypic behavior.	Javadfar et al. [132]. Randomized, double-blind, placebo-controlled, parallel-group trial
Vitamin D3	2000 IU/day	117 children with ASD (age: 2.5–8.0 years)	The rate of positive response (at least a 25% reduction in the ABC-hyperactivity score and the ABC-irritability score) was 68% and 63%, respectively.	Mazahery et al. [133]. 12-month randomized double-blind, placebo-controlled study
Vitamin D3	Administered IM at 150,000 IU/month (a total of three injections) and orally at 400 IU per day for 3 months	215 children with ASD (mean age: 4.76 ± 0.95 years (37 autistic children received vitamin D3)	Serum levels of 25(OH) D were negatively correlated with total ABC scores and language subscale scores. Vitamin D3 supplementation significantly decreased CARS and ABC symptom scores. The treatment effects were more pronounced in younger children with ASD (≤3 years).	Feng et al. [134].
Vitamin D3	2000 IU for 20 weeks	42 children with ASD	The primary endpoint (stereotypical behavior subscale of the ABC) did not show any observable effect. The self-care score in the DD-CGAS improved in the D3 group (*p* = 0.02). There was a tendency toward reduced inappropriate speech in the placebo group compared to the D3 group (*p* = 0.08), although the difference was not significant.	Kerley et al. [135]. Parallel, randomized, double-blind, placebo-controlled trial involving two visits to a clinic
Omega -3–6-9	706 mg of omega-3 fatty acids (including 338 mg of EPA and 225 mg of DHA), 280 mg of omega-6 fatty acids (including 83 mg of GLA), and 306 mg of omega-9 fatty acids	31 children aged 18–38 months	The magnitude of clinical benefit was moderate for anxious and depressed behaviors (*p* = 0.049) and internalizing behaviors (*p* = 0.05), and large for adaptive behaviors in interpersonal relationships (*p* = 0.01). The remaining behaviors and sleep were not affected.	Boone et al. [136]. 90-day randomized (1:1), double-blinded, placebo-controlled trial
Omega-3	706 mg of omega-3 fatty acids (including 338 mg of EPA and 225 mg of DHA), 280 mg of omega-6 fatty acids (including 83 mg of GLA), and 306 mg of omega-9 fatty acids	31 children (age: 18–38 months)	Greater reduction in ASD symptoms on the Brief Infant Toddler Social Emotional Assessment ASD scale. No other outcome measure reflected a similar magnitude or a significant effect.	Keim et al. [137]. 90-day randomized, fully blinded, placebo-controlled trial
Omega-3	1000 mg of omega-3 daily in the experimental group and 1000 mg of medium-chain triglycerides (placebo) daily in the control group for 8 weeks	54 children, (age: 5–15 years)	Significant improvement in stereotyped behaviors (*p* = 0.02), social communication (*p* = 0.02), and the GARS score (*p* = 0.001). No significant change in scores on the social interaction subscale.	Doaei et al. [138]. A double-blind, randomized clinical trial

## Data Availability

Not applicable.

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
