# Peer review of "A Review of the Nutritional Approach and the Role of Dietary Components in Children with Autism Spectrum Disorders in Light of the Latest Scientific Research"

_nutrients, 2023, doi:10.3390/nu15234852_

Round 1

Reviewer 1 Report

Comments and Suggestions for Authors

The authors have done a meticulous job bringing to readers' attention a broad overview of the impact of nutritional approaches and the role of specific dietary components in children with autism spectrum disorders. The structure is well organized with a division into chapters and paragraphs that makes it easy to read. Tables also functionally support the reading. The conclusions well specify what the strengths and limitations of this review are. Below are some comments and changes that the authors should make:

Line 43: is the first time the acronym CDC appears, please write in full

Line 60: Mitochondria play a key role in oxidative stress and in maintaining the cellular redox state. Dietary supplementation with a formulation nearly overlapping with nutritional factors supporting the metabolic pathways providing GSH and SAM has been shown to improve hepatic mitochondrial function in mice with diet-induced obesity. The authors might consider supplementing this aspect. Below are some useful references: 10.3390/ijms22062862; 10.1016/B978-0-12-811752-1.00004-3

Line 92: Chapter 2 has only one paragraph? I would consider changing the title of the chapter (e.g., Selective eating as a nutritional problem).

Table 2: RDA and DRI it is not clear what these acronyms mean. Is the first time it appears.

Line 204: Beginning the paragraph in this way gives the impression that despite a gluten-free or casein-free diet, ASD children have urinary excretion of exorphins from dietary sources containing gluten and casein.

Line 416: Please specify that the BTBR strain is an animal model of ASD. Moreover, it could be interesting to report studies conducted in this animal model, highlighting the gut dysbiosis, mitochondrial dysfunction and inflammatory profile founded in animal model of ASD. doi: 10.1038/srep45356; https://doi.org/10.1016/j.cell.2019.05.004.

Author Response

Dear Reviewer,

Thank you for reviewing our article. Your feedback and insights have been immensely helpful in improving the quality of the manuscript. We appreciate the time and effort dedicated to reviewing our work. Our detailed responses are presented below.

The authors have done a meticulous job bringing to readers' attention a broad overview of the impact of nutritional approaches and the role of specific dietary components in children with autism spectrum disorders. The structure is well organized with a division into chapters and paragraphs that makes it easy to read. Tables also functionally support the reading. The conclusions well specify what the strengths and limitations of this review are. Below are some comments and changes that the authors should make:

  • Line 43: is the first time the acronym CDC appears, please write in full

The acronym CDC was written in full upon first use.

  • Line 60: Mitochondria play a key role in oxidative stress and in maintaining the cellular redox state. Dietary supplementation with a formulation nearly overlapping with nutritional factors supporting the metabolic pathways providing GSH and SAM has been shown to improve hepatic mitochondrial function in mice with diet-induced obesity. The authors might consider supplementing this aspect. Below are some useful references: 3390/ijms22062862; 10.1016/B978-0-12-811752-1.00004-3

Thank you very much for this suggestion. The provided reference supports this argument.

  • Line 92: Chapter 2 has only one paragraph? I would consider changing the title of the chapter (e.g., Selective eating as a nutritional problem).

As suggested, the title of the entire chapter was changed to “Selective eating as a nutritional problem” because it addresses a single issue.

  • Table 2: RDA and DRI it is not clear what these acronyms mean. Is the first time it appears.

These acronyms were written in full under Table 2.

  • Line 204: Beginning the paragraph in this way gives the impression that despite a gluten-free or casein-free diet, ASD children have urinary excretion of exorphins from dietary sources containing gluten and casein.

The sentence 'A gluten-free and/or casein-free diet (GFCF) is the most common restrictive diet in individuals with ASD' was removed to avoid ambiguity.

  • Line 416: Please specify that the BTBR strain is an animal model of ASD. Moreover, it could be interesting to report studies conducted in this animal model, highlighting the gut dysbiosis, mitochondrial dysfunction and inflammatory profile founded in animal model of ASD. doi: 10.1038/srep45356; https://doi.org/10.1016/j.cell.2019.05.004.

Thank you very much for this comment. The information that BTBR mice are commonly used as an animal model of ASD was included in the manuscript. Studies conducted on the BTBR strain were discussed in the context of research on the microbiome-gut-brain axis in subsection 3.2.

Reviewer 2 Report

Comments and Suggestions for Authors

A review of the role of dietary components in children with ASD is an exciting topic. The authors tried to discuss the role of different diets, several nutritional elements, and dietary supplements in ASD children. However, the review requires some organized writing and well-structured Tables. Here are the comments:

Fig 1 missing.

Suggestion numbering the table in the text should be Table 1, Table 2, and so on.

Numbering the supplementary table should be Table S1, Table 2, and so on

But in this review, Table 1, Table 2, and Table 4 instead of Table 3. Similarly, S3 instead of Table S1 and Table S8 instead of Table S2, and the mixing numbers with the main text table and supplementary tables could be more precise.

Table 4 Animal studies study group the age of animals whether they are equivalent to children as the review title mentioned “role of dietary components in children with autism spectrum disorders.”

In Human studies, whether 45 children aged 3–8 years & 15 children aged 2–17 years have ASD?

Six ASD patients – Are they children? There is no mention of it. Why is a child with autism and epilepsy and a 6-year-old child included?   

There is a redundancy in the table and the text.

Table 5. Human studies - It does not look like a table format.

Table 6. There is no need for a table, and it can be written in text.

Page 15, line 416: What are BTBR mice?

The supplements section and Tabe 7 should be concise and clear.

In sum, there are several redundancies in writing and making tables. The manuscript review requires significant revisions and should be concise.  

Author Response

Dear Reviewer,

Thank you for reviewing our article. Your feedback and insights have been immensely helpful in improving the quality of the manuscript. We appreciate the time and effort dedicated to reviewing our work. We look forward to incorporating your suggestions to make the paper even stronger. Our detailed responses are presented below.

A review of the role of dietary components in children with ASD is an exciting topic. The authors tried to discuss the role of different diets, several nutritional elements, and dietary supplements in ASD children. However, the review requires some organized writing and well-structured Tables. Here are the comments:

  • Fig 1

We apologize for this oversight. Figure 1 lists the common side-effects of nutritional therapies in autism, and it has been added to the revised manuscript.

  • Suggestion numbering the table in the text should be Table 1, Table 2, and so on.

Numbering the supplementary table should be Table S1, Table 2, and so on

But in this review, Table 1, Table 2, and Table 4 instead of Table 3. Similarly, S3 instead of Table S1 and Table S8 instead of Table S2, and the mixing numbers with the main text table and supplementary tables could be more precise.

Thank you for this suggestion. Indeed, table numbering could be misleading, and it was modified.

  • Table 4Animal studies study group the age of animals whether they are equivalent to children as the review title mentioned “role of dietary components in children with autism spectrum disorders.”

Thank you for this suggestion, but in our opinion, animal testing is an indispensable part of scientific experimentation. The so-called 'model organisms' are characterized by genetic and physiological similarities. Animal models are essential in research studies aiming to elucidate the pathogenesis of diseases. There is no known cause of autism spectrum disorders (ASD), and patients are diagnosed based solely on behavioral symptoms. Furthermore, individuals with ASD have numerous genetic, neuroanatomical and immunological abnormalities, which supports the development of various animal models of ASD [Ptaszek et al. 2015. Can the use of animal models be helpful in explaining the causes of autism spectrum disorders? Issues currently addressed by Young Scientists. 1. 73-78.]

  • In Human studies, whether 45 children aged 3-8 years & 15 children aged 2-17 years have ASD?

Yes, all children in the referenced studies had ASD.

  • Six ASD patients - Are they children?

Yes, these patients were children aged 4-14 years.

  • Why is a child with autism and epilepsy

Thank you for this observation. This reference was removed because the cited study analyzed comorbidities that could have distorted the results.

  • and a 6-year-old child included?

Patients adhering to the KD were rarely studied, which is why we included a case study in the review.

  • Table 5.Human studies - It does not look like a table format.

Thank you for this remark. Table numbers were changed in the revised manuscript, and the former Table 5 is now listed as Table 4. The formatting was not changed because in our opinion, data are more effectively summarized in tabular format. A description of Table 4 was added in the text.

  • Table 6.There is no need for a table, and it can be written in text.

Thank you for this suggestion. The foods recommended in the Specific Carbohydrate Diet were described in the text.

  • Page 15, line 416:What are BTBR mice?

BTBR mice naturally exhibit autism-like behaviors and are widely used in research as an animal model of autism. Research on BTBR mice is yielding promising results that can be extrapolated to human studies. Additional information was provided in subsection 3.2.

  • The supplements section and Table 7 should be concise and clear.

Subsection 3.2 was simplified and restructured. In our opinion, Table 7 (changed to Table 6 in the revised manuscript) presents the results of the reviewed studies in a clear and concise manner. These results were not additionally described in the text due to space constraints.

  • In sum, there are several redundancies in writing and making tables. The manuscript review requires significant revisions and should be concise.  

We are grateful for the valuable comments which have improved the quality of our manuscript. The purpose of the manuscript was to aggregate and discuss the current state of knowledge on the role of diet and nutrition in ASD, and tables enable readers to quickly find the sources of the presented information. The manuscript has been revised according to the comments made by all reviewers, and we hope it is a reliable source of information.

Reviewer 3 Report

Comments and Suggestions for Authors

The article is devoted to an actual problem of the nutritional approach and the role of diet in children with autism. Technically, the manuscript is written in good English scientific language and fully reveals the declared subject. I have marked the only sentence that need editing with a highlighter.

P. 12 L. 285-288 – consider revision, the sizes of samples and controls are unclear.

I would also like to address a more general comment to the authors: ASD is currently diagnosed according to behavioral criteria that overlook clinical, pathophysiological, and genomic heterogeneity, thus repeatedly resulting in failed clinical and dietary trials. Since a completely individual selection of diet and supplementation for each patient looks utopian, the authors need to emphasize the importance of subtyping of ASD patient with a similar underlying pathophysiology (Nordahl CW, Andrews DS, Dwyer P, Waizbard-Bartov E, Restrepo B, Lee JK, Heath B, Saron C, Rivera SM, Solomon M, Ashwood P, Amaral DG. The Autism Phenome Project: Toward Identifying Clinically Meaningful Subgroups of Autism. Front Neurosci. 2022 Jan 17;15:786220. doi: 10.3389/fnins.2021.786220.)

Comments on the Quality of English Language

Technically, the manuscript is written in good English scientific language and fully reveals the declared subject. I have marked the only sentence that need editing with a highlighter.

Author Response

Dear Reviewer,

Thank you for reviewing our article. We are grateful for your feedback, and we appreciate and the time and effort devoted to evaluating our work.  

The article is devoted to an actual problem of the nutritional approach and the role of diet in children with autism. Technically, the manuscript is written in good English scientific language and fully reveals the declared subject. I have marked the only sentence that need editing with a highlighter.

  • 12 L. 285-288 – consider revision, the sizes of samples and controls are unclear.

Thank you for this suggestion. We agree that this fragment was not clearly described; therefore, it was edited and supplemented in the revised manuscript.

  • I would also like to address a more general comment to the authors: ASD is currently diagnosed according to behavioral criteria that overlook clinical, pathophysiological, and genomic heterogeneity, thus repeatedly resulting in failed clinical and dietary trials. Since a completely individual selection of diet and supplementation for each patient looks utopian, the authors need to emphasize the importance of subtyping of ASD patient with a similar underlying pathophysiology (Nordahl CW, Andrews DS, Dwyer P, Waizbard-Bartov E, Restrepo B, Lee JK, Heath B, Saron C, Rivera SM, Solomon M, Ashwood P, Amaral DG. The Autism Phenome Project: Toward Identifying Clinically Meaningful Subgroups of Autism. Front Neurosci. 2022 Jan 17;15:786220. doi: 10.3389/fnins.2021.786220.)

Thank you for this suggestion, Indeed, dietary interventions cannot be tailored to patients' individual needs, and general principles do not take into account the specific behaviors that influence the nutritional and metabolic capabilities of people with ASD. In the revised manuscript, the Conclusions section was expanded to indicate that autism is a heterogeneous condition that poses a challenge for health care providers and parents of children with ASD.